# The Association between Baseline Proton Pump Inhibitors, Immune Checkpoint Inhibitors, and Chemotherapy: A Systematic Review with Network Meta-Analysis

**DOI:** 10.3390/cancers15010284

**Published:** 2022-12-31

**Authors:** Yu Chang, Wan-Ying Lin, Yu-Cheng Chang, Chin-Hsuan Huang, Huey-En Tzeng, Eahab Abdul-Lattif, Tsu-Hsien Wang, Tzu-Hsuan Tseng, Yi-No Kang, Kuan-Yu Chi

**Affiliations:** 1Section of Neurosurgery, Department of Surgery, National Cheng Kung University Hospital, College of Medicine, National Cheng Kung University, Tainan 701401, Taiwan; 2Department of Family Medicine, Taipei Medical University Hospital, Taipei 100229, Taiwan; 3Department of Education, Center for Evidence-Based Medicine, Taipei Medical University Hospital, Taipei 100229, Taiwan; 4Division of Hematology and Oncology, Department of Internal Medicine, Taipei Medical University Hospital, Taipei 100229, Taiwan; 5Department of Internal Medicine, Lower Bucks Hospital, Bristol, PA 19007, USA; 6Department of Family Medicine, Wan Fang Hospital, Taipei Medical University, Taipei 106339, Taiwan; 7Cochrane Taiwan, Taipei Medical University, Taipei 106339, Taiwan; 8Evidence-Based Medicine Center, Wan Fang Hospital, Taipei Medical University, Taipei 100229, Taiwan; 9Department of Internal Medicine, Taipei Medical University Hospital, Taipei 100229, Taiwan

**Keywords:** proton pump inhibitors, immune checkpoint inhibitors, chemotherapy

## Abstract

**Simple Summary:**

Proton pump inhibitors (PPIs) are the most commonly used gastric acid suppressants in cancer patients. However, long-term use of PPIs can cause dysbiosis effects disrupting gut microbiota with subsequent impairment of the effectiveness of immune checkpoint inhibitors (ICIs). Our study demonstrates that, in advanced non-small cell lung cancer and urothelial carcinoma, patients receiving ICIs with concomitant PPIs are associated with poorer survival outcomes, when compared not only with those without PPIs but also with patients treated with chemotherapy, implying that PPIs could compromise the effectiveness of ICIs, causing them to be less effective than chemotherapy. As a result, we suggest that clinicians should avoid unnecessary PPI prescription in these patients. Conversely, there is little survival association with PPI in patients with melanoma, renal cell carcinoma, hepatocellular carcinoma, and squamous cell carcinoma of the neck and head. Nevertheless, future high quality prospective studies including more cancer types are warranted.

**Abstract:**

(1) Although emerging evidence suggests that proton pump inhibitor (PPI)-induced dysbiosis negatively alters treatment response to immune checkpoint inhibitors (ICIs) in cancer patients, no study systematically investigates the association between PPIs, ICIs, and chemotherapy; (2) Cochrane Library, Embase, Medline, and PubMed were searched from inception to 20 May 2022, to identify relevant studies involving patients receiving ICIs or chemotherapy and reporting survival outcome between PPI users and non-users. Survival outcomes included overall survival (OS) and progression-free survival (PFS). Network meta-analyses were performed using random-effects models. *p*-scores, with a value between 0 and 1, were calculated to quantify the treatment ranking, with a higher score suggesting a higher probability of greater effectiveness. We also conducted pairwise meta-analyses of observational studies to complement our network meta-analysis; (3) We identified 62 studies involving 26,484 patients (PPI = 8834; non-PPI = 17,650), including non-small cell lung cancer (NSCLC), urothelial carcinoma (UC), melanoma, renal cell carcinoma (RCC), hepatocellular carcinoma (HCC), and squamous cell carcinoma (SCC) of the neck and head. Eight post-hoc analyses from 18 randomized–controlled trials were included in our network, which demonstrated that, in advanced NSCLC and UC, patients under ICI treatment with concomitant PPI (*p*-score: 0.2016) are associated with both poorer OS (HR, 1.49; 95% CI, 1.37 to 1.67) and poorer PFS (HR, 1.41; 95% CI, 1.25 to 1.61) than those without PPIs (*p*-score: 1.000). Patients under ICI treatment with concomitant PPI also had poorer OS (HR, 1.18; 95% CI, 1.07 to 1.31) and poorer PFS (HR, 1.30; 95% CI, 1.14 to 1.48) in comparison with those receiving chemotherapy (*p*-score: 0.6664), implying that PPIs may compromise ICI’s effectiveness, making it less effective than chemotherapy. Our pairwise meta-analyses also supported this association. Conversely, PPI has little effect on patients with advanced melanoma, RCC, HCC, and SCC of the neck and head who were treated with ICIs; (4) “PPI-induced dysbiosis” serves as a significant modifier of treatment response in both advanced NSCLC and UC that are treated with ICIs, compromising the effectiveness of ICIs to be less than that of chemotherapy. Thus, clinicians should avoid unnecessary PPI prescription in these patients. “PPI-induced dysbiosis”, on the other hand, does not alter the treatment response to ICIs in advanced melanoma, RCC, HCC, and SCC of the head and neck.

## 1. Introduction

Although clinical trials have shown that immune checkpoint inhibitors (ICIs) provide significant efficacy over non-ICIs comparators in many advanced cancers reshaping their therapeutic landscape [1,2,3,4], only a certain proportion of cancer patients have demonstrated a meaningful treatment response to ICIs. Considering metastatic castration-resistant prostate cancer as an example, pembrolizumab, the only FDA approved ICI for prostate cancer, is used predominantly in patients with high microsatellite instability (MSI-H) [5]. Therefore, identifying prognostic predictors and biomarkers for treatment response in patients taking ICIs is crucial, as it helps determine the subgroups of cancer patients who would benefit most from ICIs, improving the precision of therapeutic management. In fact, a variety of clinicopathological features, including age, gender, the Eastern Cooperative Oncology Group (ECOG) performance status, tumor mutation burden, and programmed death ligand 1 (PD-L1) expression, have been discovered as potential modifiers of treatment response in patients treated with ICIs [6,7,8]. Concomitant use of various medications has also been investigated to determine whether they alter the effectiveness of ICIs [9,10]. Notably, PPIs, commonly prescribed gastric acid suppressants for cancer patients, are well-known for their dysbiosis effects that disrupts gut microbiota, which theoretically impairs patients’ response to both ICIs and chemotherapy [11,12].

Six meta-analyses [13,14,15,16,17,18] have investigated the survival impacts of concomitant PPI on the effectiveness of ICI, but their results are contradictory. One [14] demonstrated negligible survival influence of PPI on ICI. Another three [15,17,18] alluded that PPI could negatively affect ICI in advanced cancers, while the remaining two syntheses [13,16] implied melanoma patients may derive survival benefits due to lower disease recurrence. Despite these six meta-analyses, there has been little investigation and discussion on the interaction of PPI against ICI and chemotherapy. Although a subgroup analysis of trials of IMvigor 211 [19] and IMpower150 [20] showed that PPIs significantly compromised the benefit of ICIs over chemotherapy, another exploratory analysis of OAK [21] and POPLAR [22] demonstrated insignificant interaction.

Given the significant increased indications of ICIs due to actively broadening cancer types and the extensive use of PPIs, an updated synthesis is urgently warranted. Thus, our study contains two main objectives; firstly, we aim to investigate the association among PPIs, ICIs, and chemotherapy by conducting a network meta-analysis (NMA); secondly, we will update the comparative survival outcomes of PPI users and non-users in patients receiving ICIs and also in patients taking chemotherapy with a wider range of cancers.

## 2. Methods

We performed the present systematic review and meta-analysis based on the Cochrane Handbook for Systematic Reviews of Interventions [23] and reported results in accordance with the Preferred Reporting Items for Systematic Reviews and Meta-Analyses (PRISMA) Network Meta-analysis extension statement, and Meta-analysis Of Observational Studies in Epidemiology guidelines (Appendix A). The study was registered on PROSPERO (CRD 42021258800).

### 2.1. Study Selection

PubMed, Medline, Embase, and Cochrane Library were searched, from inception until 20 May 2022. Three investigators (W.Y.L, Y.C, and Y.C.C) independently identified relevant studies, and discrepancies were addressed by reaching a consensus with senior reviewers (K.Y.C and Y.N.K). Search details are presented in Appendix A.

### 2.2. Eligibility Criteria

Three predefined criteria for evidence selection were as follows: (1) randomized–controlled trials (RCTs), prospective, or retrospective cohort studies; (2) studies involving adult patients aged over 18 with cancers receiving ICIs or chemotherapy without concurrent radiotherapy; (3) studies reporting at least one comparative survival outcome between PPI users and non-users, overall survival (OS), or progression-free survival (PFS) (PPI versus non-PPI users), irrespective of indications.

### 2.3. Data Extraction

Three investigators (H.C.C., E.A, and T.H.W) independently extracted relevant information from eligible articles. Details are available in Appendix A.

### 2.4. Quality Assessment

Two investigators (T.H.T and K.Y.C) independently completed a critical appraisal of the included literature by using the Cochrane Risk of Bias tool 2.0 for RCTs, and the Risk of Bias in Non-randomised Studies–of Interventions (ROBINS-I) tool for non-RCTs. Any discrepancy was addressed through discussion with the third investigator (Y.N.K).

### 2.5. Assessment of Transitivity Assumption

A pivotal concept of NMA is that patients in the network are jointly randomizable. One plausible assessment of transitivity assumption is to place included trials under scrutiny to examine whether important effect modifiers are similarly distributed throughout the network [24]. We pre-specified age, gender, ECOG performance status, and PDL-1 expression status as effect modifiers since these factors are known prognostic factors for cancer patients.

### 2.6. Main Outcomes and Statistical Analysis

OS and PFS were pooled by obtaining the unadjusted hazard ratio (HR) extracted directly from each reference. When studies did not report the HR but presented Kaplan–Meier survival curves instead, we acquired an estimated HR from the curves through a well-established method [25], i.e., by using a calculation spreadsheet developed by Tierney and colleagues [26]. For baseline effect modifiers, we used weighted mean difference (WMD) and risk ratio (RR) through an inverse variance method to pool continuous and binary characteristics, respectively. When continuous outcomes are not reported as mean with standard difference but instead median with interquartile or range format, which cannot be used for quantitative syntheses, we utilized a well-established and well-validated tool [27] to convert the data to an appropriate format for syntheses. All estimated effects were presented with a 95% confidence interval (CI).

Network and head-to-head meta-analysis were conducted using RStudio with the ‘netmeta’ and ‘meta’ package, respectively (Appendix A). For the NMA, we produced a network graph to illustrate the structure of evidence followed by league tables for summary of frequentist NMA using a random-effect model. Regarding the league table, interventions were ranked by their *p*-scores with the netrank function; *p*-scores were a value between 0 and 1, with a higher score suggesting a higher probability of greater effectiveness. Forest plots of HRs and 95% CIs were generated with “chemotherapy” as the reference. Inconsistency was evaluated through the netsplit function https://jamanetwork.com/journals/jamanetworkopen/fullarticle/2773396, accessed on 6 June 2022—zoi200800r45 and displayed via heat plots with the netheat command. We also performed sensitivity analyses by excluding trials of potential source of inconsistency, unknown PDL-1, and treatment line. Function ‘netmeta::funnel’ was used for depicting the comparison-adjusted funnel plot, which is a common method for depicting publication bias in NMA.

For pairwise meta-analysis, the pooled estimate was based on random-effects with the restricted maximum likelihood method due to inevitable between-trial variance. Heterogeneity was assessed using I-square [28], with values of I^2^ < 25%, 25% < I^2^ <50%, and I^2^ > 50% indicating low, moderate, and high heterogeneity, respectively. Pre-specified sensitivity analyses include subgroup analyses based on cancer types as well as treatment line, and exclusion of studies subject to high risk of bias. Determination of statistical significance in these analyses followed common threshold (*p* < 0.05). Additionally, function ‘funnel’ and ‘metabias’ are used for examining publication bias when a head-to-head meta-analysis with more than 10 studies.

## 3. Results

The study identified 21,059 references, with 102 studies for full-text inspection, among which 40 studies did not meet the eligibility criteria (Appendix A). Eventually, we included a total of 62 studies [9,10,19,20,21,22,29,30,31,32,33,34,35,36,37,38,39,40,41,42,43,44,45,46,47,48,49,50,51,52,53,54,55,56,57,58,59,60,61,62,63,64,65,66,67,68,69,70,71,72,73,74,75,76,77,78,79,80,81,82,83] for qualitative and quantitative syntheses (Figure 1).

### 3.1. Characteristics of Included Studies

Table 1 demonstrates there are thirty-six retrospective studies [9,10,29,30,31,34,37,38,40,41,42,47,49,50,52,55,56,57,58,59,60,61,62,64,65,67,68,69,70,72,74,75,76,77,79,80] and eight were post hoc analyses [35,36,44,45,46,53,54,78] from 18 RCTs [19,20,21,22,32,33,39,43,48,51,63,66,71,73,83,84,85,86,87,88,89,90,91], involving 26,484 patients (PPI = 8834; non-PPI = 17,650) enrolled between 2000 and 2022.

Eligibility criteria of 18 RCTs [19,20,21,22,32,33,39,43,48,51,63,66,71,73,81,82,83] are elaborated in Appendix A; notably, patients with pregnancy, known CNS metastases, clinically significant cardiovascular disease, bleeding events, coagulopathy, and the use of anticoagulants or antiplatelets were excluded.

There were 42 studies [9,10,19,20,21,22,29,30,31,32,34,37,38,40,41,42,47,49,50,51,52,55,56,57,58,59,60,61,62,65,67,68,69,71,74,75,76,77,79,80,81,82,83] which primarily investigated cancer patients receiving ICI, among which 17 studies [20,21,22,31,34,37,38,47,51,56,61,65,67,69,71,74,75], 8 studies [19,32,58,62,68,76,79,80], 7 studies [29,40,41,59,75,81,82,83], 3 studies, 2 studies [52,55], 1 study, and 10 studies [9,10,30,41,42,49,50,59,60,75] reported survival outcomes of advanced non-small cell lung cancer (NSCLC), urothelial carcinoma (UC), melanoma, renal cell carcinoma (RCC), hepatocellular carcinoma (HCC), squamous cell carcinoma (SCC) of the head and neck, and uncategorized cancer cohorts, respectively. Information on uncategorized cancers is listed on Appendix A. There were 11 studies (retrospective: 4 [64,70,72,77] and post hoc analysis: 7 [33,39,43,48,63,66,73]) which investigated the effect of PPIs on patients with colorectal cancer (CRC) receiving chemotherapy. Prespecified effect modifiers, including age, sex, ECOG, and PDL expression are comparable between PPI users and non-users across the network (Appendix A). Details of PPI are presented in Appendix A. The sources of risk of bias arise mostly from bias due to confounding (Appendix A). No study was evaluated as critical risk of bias. Appendix A provides detailed elaboration of ROBINS-I of each domain.

### 3.2. Network Meta-Analysis

Eight post hoc analysis [35,36,44,45,46,53,54,78] of 18 RCTs [19,20,21,22,32,33,39,43,48,51,63,66,71,73,81,82,83] enrolling advanced cancers (advanced NSCLC, UC, melanoma, CRC, and gastroesophageal adenocarcinoma) are included in the network. Through the visualization of network plot (Figure 2), patients under ICI treatment with concomitant PPI (red node) are associated with poor OS and PFS, compared not only with those under ICI without PPI, but also with those with chemotherapy. Details of NMA are presented in Appendix A. However, both the netsplit and netheat plots demonstrated significant inconsistency throughout the comparisons (Appendix A). After excluding trials of melanoma (Appendix A), NMA also indicates that patients under ICI treatment with concomitant PPI are associated with poor OS (HR, 1.49; 95% CI, 1.37 to 1.67) and PFS (HR, 1.41; 95% CI, 1.25 to 1.61), compared with those without PPI, and with worse OS (HR, 1.18; 95% CI, 1.07 to 1.31) and PFS (HR, 1.30; 95% CI, 1.14 to 1.48), compared with those with chemotherapy (Appendix A), with a substantial decrease in inconsistency (Appendix A). According to the net league table with *p*-scores, ICI without PPI is ranked highest, followed by chemotherapy without PPI, ICI with PPI, and chemotherapy with PPI (Table 2). The funnel plot (Appendix A) demonstrated general symmetry through inspection, which was further supported by the Egger’s test (*p* = 0.71), indicating no publication bias from small study effects.

### 3.3. Prespecified Sensitivity Analyses (Appendix A)

We excluded data of IMpower 130, IMpower 131 [51], IMvigor 210, TRIO-013/LOGiC [43], AVF2107g, N016966, and Carrato 2013 [33] because these trials enrolled treatment-naïve patients (Appendix A). Regarding the PDL-1 expression, we excluded data of Chu 2017 [36] and Kichenadasse 2021 [53] because of unavailable PDL-1 information (Appendix A). After the sensitivity analyses, the association among PPI, ICI, and chemotherapy remains the same, with no significant inconsistency.

### 3.4. Pairwise Meta-Analysis for ICI Cohorts (Figure 3 and Appendix A)

In patients with NSCLC, the use of PPIs poses a 36% higher risk of death (HR, 1.36; 95% CI, 1.28 to 1.45; I^2^ = 0%) and disease progression (HR, 1.36; 95% CI, 1.26 to 1.47; I^2^ = 0%) than those without PPIs. The same trend is observed in patients with UC regarding OS (HR, 1.71; 95% CI, 1.49 to 1.96; I^2^ = 0%) and PFS (HR, 1.55; 95% CI, 1.37 to 1.75; I^2^ = 0%). There is no significant difference in either of the survival outcomes with respect to advanced melanoma, RCC, HCC, and SCC of the head and neck. Scatters in the funnel plot with Egger’s test suggest no publication bias in either OS or PFS (Appendix A). Sensitivity analysis of excluding studies subject to high risk of bias yields the same association with the significant decrease in the statistical heterogeneity (Appendix A).

### 3.5. Pairwise Meta-Analysis for Chemotherapy Cohorts (Appendix A)

The use of PPIs is associated with a significant 12% higher all-cause mortality in patients with advanced NSCLC receiving chemotherapy, with a marginally significant higher progression rate (Appendix A). For GI cancers, using a PPI also confers a significantly higher risk of death and disease progression rate (Appendix A) but with substantial heterogeneity.

When CRC cancer patients are stratified based on chemotherapeutic regimens, concomitant PPI use negatively affects the effectiveness of these regimens in patients with FU-based regimens (Appendix A). Conversely, survival outcomes are similar between PPI users and non-users in patients receiving capecitabine-based regimens (Appendix A).

### 3.6. Meta-Analysis Using Adjusted HR (Appendix A)

Appendix A provides adjusted variables for adjusted outcomes and Appendix A provide the results of the meta-analysis using adjusted HR. No significant discrepancy is noted between unadjusted and adjusted outcomes. We finally summarize the meta-analyses findings with both unadjusted and adjusted results in Appendix A.

## 4. Discussion

Although various meta-analyses have investigated the influence of PPIs on the effectiveness of ICIs, no synthesis, to date, has explored the interaction among PPIs, ICIs, and chemotherapy. Our NMA demonstrated that baseline PPI use not only has a negative prognostic influence on advanced cancer patients treated with ICIs but is shown to compromise the effectiveness of ICI, causing it to be even worse than chemotherapy. It is also noteworthy that trials in the network not only included advanced cancer patients with ECOG PS < 2 but explicitly excluded patients with medical conditions warranting long-term antiplatelets or anticoagulants that may require baseline PPI to prevent GI bleeding, such as coronary artery disease and thromboembolism, which reflect high burdens of co-morbidities and which can obviously confound PPI’s survival impact. Therefore, there is a low risk of unmeasured confounding bias for our network, although included trials contain no pertinent information on the indications of PPI. However, through the visualization of netheat plots (Appendix A), significant inconsistency exists in the network, with melanoma deemed to be the source of the inconsistency, which decreased substantially after the removal of melanoma trials. This, together with the head-to-head meta-analyses, provides a plausible reason as to why melanoma contributes to the source of inconsistency, as melanoma was shown to be neutrally affected by PPI. Conversely, only advanced NSCLC and UC patients treated with ICIs are negatively affected by PPI. Although effect modifiers were comparably distributed and no significant inconsistency was detected in our NMA after the removal of melanoma trials, conceptual heterogeneity was inevitable as different cancers encompass distinct histopathological features, clinical behaviors, and responses to therapy. Ideally, for the most precise synthesis, every cancer type should possess their own network. However, the number of trials to date are too limited for us to construct an individual network for respective cancer types. We acknowledge this as our major limitation, and the findings of our NMA should be interpreted together with pairwise meta-analyses supplemented by real-world evidence.

On the basis of our syntheses, “PPI-induced dysbiosis” serves as a significant modifier of treatment response to ICIs in both advanced NSCLC and UC, and various basic science studies have already laid a solid foundation for this clinical observation. The higher diversity of gut microbiota correlates with a higher response to ICIs owing to its positive correlation with T-cell numbers and activity [11]. Notably, PPI users were found to have lower diversity, regardless of indications, compared with non-users, with Firmicutes being the most strongly affected species [84]. Moreover, Routy et al. [11] indicated that NSCLC patients responded well to anti-PD1 agents that had an over-presentation of Firmicutes and higher memory T-helper cell reactivity against commensals in the peripheral blood. Another possible explanation for the negative association between the effectiveness of PPIs and ICIs may be related to *H. pylori* infection. Oster and colleagues [85] unveiled that *H. pylori* infection negatively altered the response to ICI in a pre-clinical setting. They also demonstrated that NSCLC patients with seropositive *H. pylori* were associated with significantly lower OS and PFS when compared with those with *H. pylori* seronegativity. Consequently, PPI could be a potential surrogate marker for *H. pylori* infection causing the negative association observed in our study. Another study also implied that dysbiosis effects were minimal when regimens were administered in a later-line context [45], due to the development of immunosurveillance evasion in advanced cancers [86]. However, we demonstrate that the detrimental effects of PPIs may be independent of the treatment line on the basis of the prespecified sensitivity analyses.

On the other hand, “PPI-induced dysbiosis” does not seem to not alter the treatment response to ICIs in patients with advanced melanoma, RCC, HCC, and SCC of the head and neck. This result contradicts previous syntheses [13,16] that demonstrated the potential benefit of PPIs for melanoma patients treated with ICIs. There are two feasible explanations for such a distinctive observation. Firstly, only two studies were included in previous syntheses, which underlies the convincing mechanism, as a small body of evidence can underpower the result, giving rise to a false positive association. Secondly, the quality assessment tool used in their syntheses was the Newcastle–Ottawa Scale, which is now considered outdated and unreliable due to the advent of ROBINS-I, which is the gold-standard tool in current evidence-based medicine. When ROBINS-I was used in the appraisal of these two studies, it was discovered that the study [40] contributing to the positive effect of PPI on melanoma was subject to high risk of bias. Note that after excluding this study [40], statistical heterogeneity decreased substantially, as we demonstrated in Appendix A. In fact, pre-clinical studies have shed much light on potential dysbiosis effects on melanoma. For those treated with ICI, responders were found to have a significantly higher alpha diversity of gut microbiota than non-responders [87,88], and using antibiotics during the treatment window was associated with a shorter PFS [89,90]. However, PPIs were shown to exert a pro-apoptotic effect on melanoma cells through the inhibition of vacuolar ATPase, an ATP-dependent proton pump [91,93], which disturbs pH gradient across melanoma cells, resulting in caspase-dependent cell death [94]. We assumed that these two aforementioned mechanisms counterbalance each other and contribute to neutral survival impacts of PPI on melanoma. For patients with advanced RCC, our result resonates with a large pooled analysis [95] of individual data from clinical trials illustrating that PPIs conferred to negligible survival influence on RCC patients treated with tyrosine kinase inhibitors (TKIs), presumably due to insignificant effects on bioavailability of TKIs. The same neutral association is observed in patients with HCC, and SCC of the head and neck as well. However, such association in these cancers currently lacks a plausible mechanism, and clinicians should keep in mind that it exhibits a false negative due to small number of studies and the small sample size. We anticipate more RCTs or large observational studies of these cancers for larger pooled analysis in the future.

Regarding the chemotherapeutic modalities, they varied in CRC among included trials. We believe that the high variance in therapeutic modalities among CRC trials accounts for the main origins of heterogeneity, not only in pairwise analysis but also in NMA. Nonetheless, our study shows that patients treated with capecitabine-based therapy are not affected by baseline PPIs. Capecitabine is almost completely metabolized to fluorouracil after being absorbed through the GI walls, and it was believed that PPI hampers the solubility, absorption, and distribution of capecitabine by increasing the intragastric pH level [96]. However, in vitro data appear not to support this interaction, as capecitabine does not become ionized until it reaches a pH level of 8.8, when it is poorly absorbed, which is obviously higher than that induced by PPI [97,98]. In contrast, fluorouracil-based agents are negatively affected by concomitant PPI, although with considerable heterogeneity. This appears to arise from complex combinations with other chemotherapeutic agents and target therapy. Unfortunately, to date, scarce pharmacokinetic studies explore the effect of PPIs on oxaliplatin, irinotecan, and target therapy, and our findings suggest that interactions between PPIs and these modalities should be addressed in the future, warranting further clinical and pharmacokinetic investigations.

One caveat of incorporating CRC into the network is that ICIs currently have limited roles in the CRC treatment field. Although a recent trial, KEYNOTE-177 [99], has established the antitumor efficacy of ICIs in treatment-naïve advanced CRC, only patients harboring deficient-MMR/MS-instable tumors are eligible for using ICIs, with the rationale being that a high mutational burden from deficient-MMR/MS-instability results in overexpression of immunogenic antigens and upregulation of immune checkpoint proteins [100]. On the other hand, the role of ICI in advanced CRC with preserved-MMR/MS-stable diseases is still under exploration, with multiple active recruiting trials investigating the combination of ICI and traditional regimens, which is hypothesized to evoke immunogenic responses.

Another limitation to our network is in regard to the PDL-1 status. Although PDL-1 expression is well-balanced across our network, and sensitivity analyses demonstrated no significant impact of PDL-1 status on our NMA, readers should bear in mind that they suffer from three limitations. Firstly, different cancers contain their own exclusive immune profile and present distinctive responses to different ICIs with a varying correlation to the PDL-1 expression. For instance, in POPLAR and OAK trials, advanced NSCLC patients with higher PDL-1 expression treated with atezolizumab derived higher survival benefits over chemotherapy [21,22] Conversely, in advanced UC, the issue of whether ICI has survival superiority over chemotherapy varied among different agents. Pembrolizumab was shown to result in better survival benefits than chemotherapy, [101,102] whereas atezolizumab did not [19]. The role of PDL-1 expression remains undetermined in UC to date, with nivolumab (CheckMate 275) [103], durvalumab [104], and pembrolizumab (KeyNote 045) [101,102] predicting a better survival response rate using PDL-1, unlike atezolizumab (IMvigor 211) which does not. Furthermore, for CRC patients, Checkmate 142 [105] suggested the PDL-1status is not a predictive biomarker, but a mismatch repair/DNA microsatellite instability (MMR/MS) is. Secondly, PDL-1 expression can be evaluated by different methodologies. There are four commonly used immunohistochemistry assays, including 22C3, 28-8, SP142, and SP263, to stain tumors, and these scoring systems were not directly comparable metrics of PDL-1 expression level. Thirdly, it should be noted that studies of unknown PDL-1 status correspond exactly to CRC studies, which are also a source of heterogeneity. Thus, it implies our sensitivity analysis could be potentially biased.

In summary, the present study contains following novelties. Firstly, we investigated the interaction of PPIs against ICIs versus chemotherapy, which has allowed us to gain further insight into the detrimental magnitude of PPIs in affected cancers. Secondly, only NSCLC and UC are negatively affected by PPI-induced dysbiosis and melanoma, whereas RCC, HCC, and SCC do not seem to be affected. Therefore, when cancer patients have indications of using gastric acid suppressants, such as antiemesis for ICI and chemotherapy, peptic ulcers, and reflux esophagitis, H_2_-blockers may be considered given PPI’s potential detrimental effects on ICI in certain cancers and on specific chemotherapeutic regimens. On the other hand, several limitations should be addressed in the future. Firstly, constructing respective networks for different cancers would provide greater knowledge of the interaction between PPIs and therapeutic modalities. Secondly, the included patients were mostly ECOG-PS 0-2, which hinders the generalizability to other cancer patients. Thirdly, information on PDL-1 expression is scarce, so we are unable to look further into the magnitude of PPIs’ effects on ICI-treated patients with different PDL-1 expressions. Last but not least, there is still inadequate data for probing the impact of PPIs on other types of malignancy, such as prostate cancer, CRC, and HCC, as ICIs are still looking for their niche in these cancers.

## 5. Conclusions

“PPI-induced dysbiosis” serves as a significant modifier of treatment response in both advanced NSCLC and UC that are treated with ICIs, compromising the effectiveness of ICIs to be less than that of chemotherapy. Thus, clinicians should avoid unnecessary PPI prescription in these patients. “PPI-induced dysbiosis”, on the other hand, does not alter the treatment response to ICIs in advanced melanoma, RCC, HCC, and SCC of the head and neck. Future high quality prospective studies including more cancer types, and more detailed PDL-1 status are warranted.

## Figures and Tables

**Figure 1 cancers-15-00284-f001:**
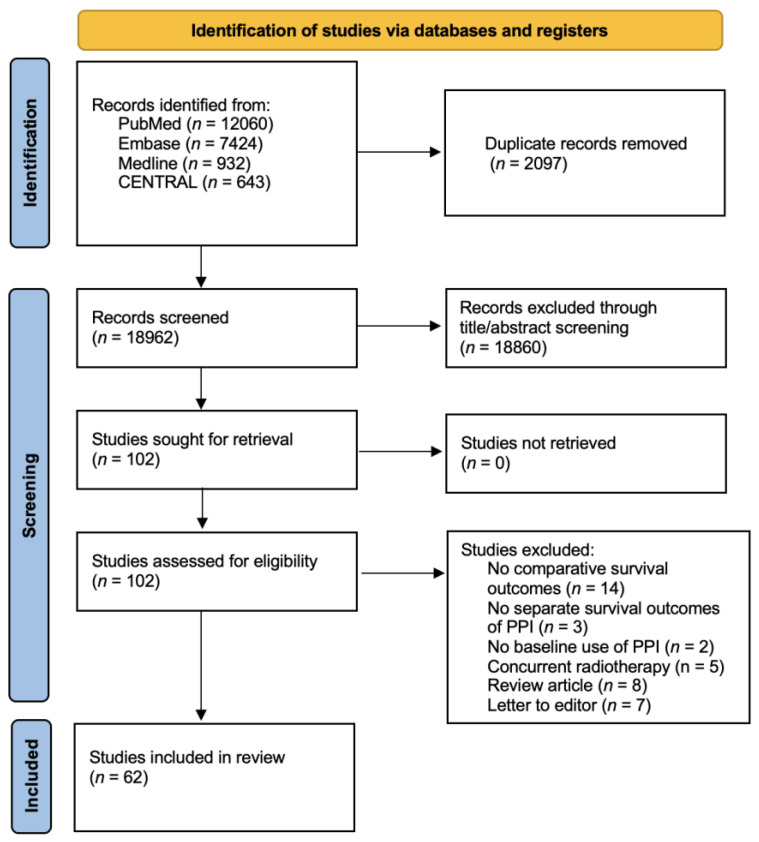
PRISMA flowchart diagram. We initially extracted a total of 21,059 potential references, including 12,060 from PubMed, 7424 from Embase, 932 from Medline, and 643 from CENTRAL. After a duplicate exclusion, 18,962 studies were identified. Screening the titles and abstracts yielded 102 full-text articles, the eligibility of which was assessed. Forty studies were excluded after reading whole texts owing to reasons elaborated in Appendix A. Eventually, 62 studies fulfilled the eligibility criteria and were included for qualitative and quantitative syntheses. PRISMA, Preferred Reporting Items for Systematic Reviews and Meta-Analyses.

**Figure 2 cancers-15-00284-f002:**
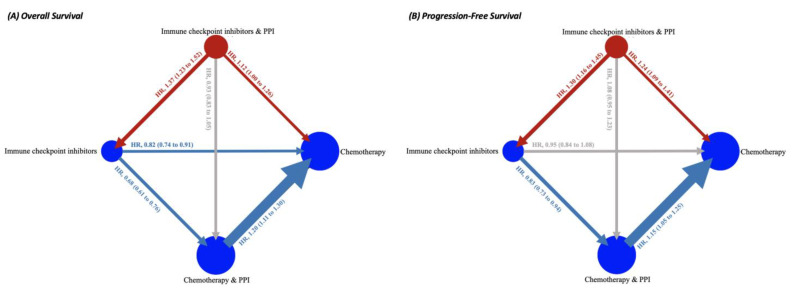
Network plot for comparative association among PPI, ICI, and chemotherapy in terms of (**A**) overall survival and (**B**) progression-free survival. The thickness of the connecting line corresponds to the number of trials among comparators. We specifically highlight the arm ICI with baseline PPI as red node, the red arrow as its significant comparison with ICI, and chemotherapy without baseline PPI to reiterate our main findings. Blue arrows also indicate significant survival association between two nodes. Conversely, gray arrows suggest little association between two arms. PPI, proton pump inhibitor; ICI, immune checkpoint inhibitor; and HR, hazard ratio.

**Figure 3 cancers-15-00284-f003:**
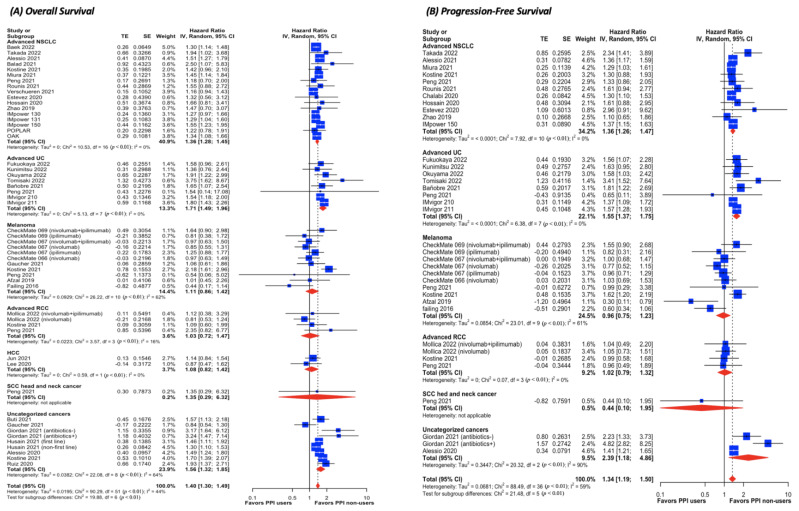
Forest plot of comparative (**A**) overall survival and (**B**) progression-free survival in cancer patients treated with ICI between PPI users and non-users. The size of squares is proportional to the weight of each study. Horizontal lines indicate the 95% CI of each study; diamond, the pooled estimate with 95%; CI, confidential interval; HCC, hepatocellular carcinoma; HR, hazard ratio; NSCLC, non-small cell lung cancer; PPI, proton pump inhibitor; RCC, renal cell carcinoma; SCC, squamous cell carcinoma; and UC, urothelial carcinoma.

**Table 1 cancers-15-00284-t001:** Study characteristics.

Included Studies	Study Type	Country	Inclusion Period	Sample Size, *n*	Therapeutic Modality			Treatment-Naïve, *n* (%)	PPI *, *n* (%)	PPI Use Window	H_2_-Blocker, *n* (%)
*Immune checkpoint inhibitors (n= 36)*					CTLA-4, *n* (%)	PD-1, *n* (%)	PD-L1, *n* (%)				
Advanced NSCLC (n = 17)											
IMpower130 [71]	RCT	8 countries	2015–2017	483	0	0	483 (100)	483 (100)	N/A¶	Following Rx	N/A
IMpower131 [51]	RCT	26 countries	2015–2017	681	0	0	681 (100)	681 (100)	N/A¶	Following Rx	N/A
IMpower150 [20]	RCT	26 countries	2015–2016	802	0	0	802 (100)	N/A	290 (36.2)	30 d before/after Rx	N/A
POPLAR [22]	RCT	USA/Europe	2013–2014	144	0	0	144 (100)	0 (0)	N/A	30 d before/after Rx	N/A
OAK [21]	RCT	31 countries	2014–2015	613	0	0	613 (100)	0 (0)	N/A	30 d before/after Rx	N/A
Baek 2022 [31]	Retrospective	South Korea	2017–2018	1646	0	1595 (97)	51 (3)	0 (0)	823 (50)	30 d before Rx	N/A
Takada 2022 [67]	Retrospective	Japan	2016–2019	95	0	85 (89.5)	10 (10.5)	N/A	37 (39)	N/A	N/A
Alessio 2021 [38]	Retrospective	Italy	2017–2020	950	0	950 (100)	0	950 (100)	474 (49.9)	N/A	N/A
Balado 2021 [34]	Retrospective	Spain	2017–2020	49	0	49 (100)	0	49 (100)	26 (53.1)	30 d before/after Rx	N/A
Kostine 2021 [75]	Retrospective	France	2015–2017	150	N/A	N/A	N/A	N/A	N/A	30 d before/after Rx	N/A
Muira 2021 [56]	Retrospective	Japan	2016–2018	300	0	300 (100)	0	40 (13.3)	163 (54.3)	N/A	N/A
Rounis 2021 [61]	Retrospective	Greece	2017~2019	66	0	66 (1000)	0	0 (0)	23 (34.8)	90 d before Rx	N/A
Verschueren 2021 [69]	Retrospective	Netherlands	2015~2019	221	0	214 (97)	7 (3)	84 (38)	96 (43.4)	30 d before/after Rx	N/A
Estevez 2020 [37]	Retrospective	Spain	2015~2018	70	0	64 (91.4)	6 (8.6)	0 (0)	59 (84.3)	90 d before Rx	N/A
Hossain 2020 [47]	Retrospective	Australia	2015~2019	63	N/A	N/A	N/A	N/A	34 (54)	28 d after Rx	N/A
Svaton 2020 [65]	Retrospective	Czech	2015~2019	224	0	224 (100)	0	9 (4.0)	64 (28.6)	30 d before/after Rx	N/A
Zhao 2019 [74]	Retrospective	China	2016–2018	109	0	109 (100)	0	28 (25.7)	40 (36.7)	30 d before/after Rx	N/A
Advanced UC (*n* = 8)											
IMvigor 210 [32]	RCT	USA	2014~2015	429	0	0	429 (100)	119 (22.5)	141 (32.9)	30 d before/after Rx	N/A
IMvigor 211 [19]	RCT	USA	2015~2017	467	0	0	467 (100)	0 (0)	145 (31.0)	30 d before/after Rx	N/A
Fukuokaya 2022 [79]	Retrospective	Japan	N/A	227	N/A	N/A	N/A	N/A	56 (24.7)	N/A	N/A
Kunimitsu 2022 [76]	Retrospective	Japan	2017~2020	79	0	0	79 (100)	0 (0)	34 (43.0)	30 d before/60 d after Rx	N/A
Okuyama 2022 [58]	Retrospective	Japan	2015~2021	155	0	0	155 (100)	0 (0)	99 (63.9)	30 d before Rx	N/A
Tomisaki 2022 [68]	Retrospective	Japan	2018–2021	40	0	40 (100)	0	0 (0)	15 (37.5)	60 d before/30 d after Rx	N/A
Bañobre 2021 [62]	Retrospective	Spain	2016~2020	119	0	39 (32.7)	80 (67.3)	22 (18.5)	54 (45.4)	N/A	N/A
Lida 2021 [80]	Retrospective	Japan	2018~2021	115	0	0	115 (100)	0 (0)	N/A	30 d before/after Rx	N/A
Advanced melanoma (*n* = 8)											
CheckMate 066 [81]	RCT	80 centers	2013~2021	210	Nivolumab: 210 (100)			210 (100)	49 (23.3)	30 d before Rx	N/A
CheckMate 067 [82]	RCT	21 countries	2013~now	945	Ipilimumab + Nivolumab: 314 (33.2); Ipilimumab: 315 (33.3); Nivolumab: 316 (33.4)			945 (100)	161 (17.0)	30 d before Rx	N/A
CheckMate 069 [83]	RCT	France/ USA	2013~2021	142	Ipilimumab + Nivolumab: 95 (67.0); Ipilimumab: 47 (33.1)			142 (100)	33 (23.3)	30 d before Rx	N/A
Gaucher 2021 [41]	Retrospective	Brazil	2010~2019	110	15 (13.6)	68 (61.8)	27 (24.6)	110 (100)	39 (35.5)	60 d after Rx	N/A
Kostine 2021 [75]	Retrospective	France	2015~2017	293	N/A	N/A	N/A	N/A	N/A	30 d before/after Rx	N/A
Peng 2021 [59]	Retrospective	USA	2014~2019	233	0	233 (100)	0	95 (40.8)	89 (38.2)	30 d before/after Rx	N/A
Afzal 2019 [29]	Retrospective	Lebanon	N/A	120	Ipilimumab and/or Pembrolizumab			N/A	29 (24.2)	N/A	N/A
Failing 2016 [40]	Retrospective	USA	2011~2014	159	Ipilimumab:159 (100)			80 (50) §	39 (24.5)	N/A	9 (6)
Advanced RCC (*n* = 3)											
Mollica 2022 [57]	Retrospective	USA	2010~2021	63	63 (100)	0	63 (100)	63 (100)	25 (39.7)	N/A	N/A
Mollica 2022 [57]	Retrospective	USA	2010~2021	156	0	0	156 (100)	110 (70.5)	88 (56.4)	N/A	N/A
Kostine 2021 [75]	Retrospective	France	2015~2017	83	N/A	N/A	N/A	N/A	N/A	30 d before/after Rx	N/A
Peng 2021 [59]	Retrospective	USA	2014~2019	233	0	233 (100)	0	95 (40.8)	89 (38.2)	30 d before/after Rx	N/A
HCC (*n* = 2)											
Jun 2020 [52]	Retrospective	USA	2017~2019	314	21 (7)	293 (93)	0	137 (43.6)	110 (35.0)	30 d before Rx	45 (14.3)
Lee 2020 [55]	Retrospective	Taiwan	2017~2019	94	N/A	N/A	N/A	N/A	30 (31.9)	30 d before Rx	N/A
Uncategorized cancers † (*n* = 10)											
Araujo 2021 [30]	Retrospective	Brazil	N/A	216	35 (16.2)	130 (60.2)	27 (12.5)	0 (0)	114 (52.8)	60 d before/after Rx	N/A
Buti 2021 [9]	Retrospective	Italy	2014~2019	217	13 (6.0)	186 (85.7)	18 (8.3)	45 (20.7)	104 (47.9)	N/A	N/A
Gaucher 2021 [41]	Retrospective	Brazil	2010~2019	370	25 (5.4)	357 (94.6)	0	87 (23.4)	149 (40.1)	60 d after Rx	N/A
Giordan 2021 [42]	Retrospective	France	2018~2019	154	0	0	154 (100)	64 (41.6)	47 (30.5)	30 d before Rx	N/A
Husain 2021 [49]	Retrospective	USA	2011~2019	1091	274 (25.1)	817 (74.9)		N/A	415 (38.0)	N/A	N/A
Peng 2021 [59]	Retrospective	USA	2014~2019	233	0	233 (100)	0	95 (40.8)	89 (38.2)	30 d before/after Rx	N/A
Alessio 2020 [10]	Retrospective	Italy	2014~2020	1012	0	956 (94.5)	56 (5.5)	396 (39.1)	491 (48.5)	N/A	56 (5.5)
Santamaria 2020 [50]	Retrospective	Spain	2015~2018	102	1 (1.0)	86 (84.3)	15 (14.7)	73 (71.6)	78 (77.2)	30 d before/after Rx	N/A
Ruiz 2020 [60]	Retrospective	Spain	from 2015	253	31 (12.3)	222 (87.7)	0 (0)	73 (28.9)	135 (53.4)	60 d before/30 d after Rx	N/A
Kostine 2021 [75]	Retrospective	France	2015~2017	635	3 (0.5)	435 (68.5)	66 (10)	N/A	293 (46.1)	30 d before/after Rx	N/A
*Chemotherapy (n = 19)*											
					Advanced NSCLC (*n* = 6)						
Verschueren 2021 [69]	Retrospective	Netherlands	2015–2019	221	Platinum-based agents			84 (38)	101 (45.7)	30 d before/after Rx	N/A
Impower 130 [71]	RCT	8 countries	2015–2017	240	Platinum-based agents			240 (100)	N/A¶	Following Rx	N/A
IMpower 131 [51]	RCT	26 countries	2015–2017	340	Platinum-based agents			340 (100)	N/A¶	Following Rx	N/A
IMpower 150 [20]	RCT	26 countries	2015–2016	400	Platinum-based agents			N/A	151 (37.8)	30 d before/after Rx	N/A
POPLAR [22]	RCT	USA/Europe	2013~2014	143		Docetaxel		0 (0)	N/A	30 d before/after Rx	N/A
OAK [21]	RCT	31 countries	2014~2015	612		Docetaxel		0 (0)	N/A	30 d before/after Rx	N/A
					Advanced UC (*n* = 1)						
IMvigor 211 [19]	RCT	USA	2015~2017	464	Platinum-based agents			0 (0)	185 (39.9)	30 d before/after Rx	N/A
					Advanced melanoma (*n* = 1)						
CheckMate 066	RCT	80 centers	2013~2021	208	Decarbazine			0 (0)	48 (23.1)	30 d before Rx	N/A
					Uncategorized cancers † (*n* = 1)						
Alessio 2021 [38]	Retrospective	Italy	2017~2020	595	Platinum-based agents			595 (100)	321 (53.7)	N/A	N/A
					FOLFOX, n (%)	FOLFIRI/ IFL, n (%)	Cape-based, n (%)				
					Gastroesophageal carcinoma (*n* = 1)						
TRIO013/ LOGiC [43]	RCT	22 countries	2008~2012	274	0	0	274 (100)	274 (100)	119 (43.4)	20% overlapping Rx	N/A
					Early-stage colorectal cancer (*n* = 3)						
Kitazume 2022 [77]	Retrospective	Japan	2009~2014	606	0	0	606 (100)	606 (100)	54 (8.9)	20% overlapping Rx	N/A
Wong 2019 [72]	Retrospective	Canada	2004~2013	389	175 (45)	0	214 (55)	389 (100)	99 (25.4)	During Rx	N/A
Sun 2016 [64]	Retrospective	Canada	2008~2012	298	0	0	298 (100) *	298 (100)	77 (26.0)	During Rx	N/A
					Advanced colorectal cancer (*n* = 8)						
Wang 2017 [70]	Retrospective	China	2010~2014	671	307 (45.8)	0	364 (54.2)	N/A	474 (70.6)	During Rx	N/A
AXEPT [73]	RCT	Asia	2013~2015	482	0	243 (50.4)	239 (49.6)	0 (0)	49 (10.2)	20% overlapping Rx	N/A
HORIZON III [92]	RCT	28 countries	2006~2009	666	666 (100)	0	0	666 (100)	87 (13.0)	During Rx	15 (2.2)
N016966 [63]	RCT	N/A	2004~2005	2035	1018 (50)	0	1017 (50)	2035 (100)	327 (32.1)	During Rx	115 (5.7)
AVF2107g [48]	RCT	3 countries	2000~2002	780	0	780 (100)	0	780 (100)	156 (20.0)	During Rx	129 (16.5)
Carrato 2013 [33]	RCT	N/A	2007~2010	348	0	348 (100)	0	348 (100)	39 (11.0)	During Rx	15 (4.2)
VELOUR [39]	RCT	28 countries	2007~2010	584	0	584 (100)	0	0 (0)	105 (18.0)	During Rx	16 (2.8)
RAISE [66]	RCT	24 countries	2010~2013	946	0	946 (100)	0	0 (0)	232 (24.5)	During Rx	36 (3.8)
*Post hoc analysis of RCTs (n = 8)*											
Hopkins 2022 [44]	Analysis of 5 trials [20,21,22,51,71]			4458	Advanced NSCLC	Immune checkpoint and chemotherapy		N/A	1225 (27.5)¶	30 d before/after Rx	N/A
Hopkins 2021 [46]	Analysis of IMpower 150 [20]			1202	Advanced NSCLC	Immune checkpoint and chemotherapy		N/A	441 (36.7)	30 d before/after Rx	N/A
Homicsko 2022 [78]	Analysis of CheckMate 066 [81]/067 [82]/069 [83]			1505	Advanced Melanoma	Immune checkpoint and chemotherapy		1505 (100)	291 (19.3)	30 d before Rx	N/A
Chalabi 2020 [35]	Analysis of OAK^7^ and POPLAR [21,22]			757	Advanced NSCLC	Immune checkpoint and chemotherapy		0 (0)	234 (30.9)	30 d before/after Rx	N/A
Hopkins 2020 [45]	Analysis of IMvigor 210 [24] and 211 [19,32]			1360	Advanced UC	Immune checkpoint and chemotherapy		119 (8.75)	471 (34.6)	30 d before/after Rx	N/A
Kim 2021 [54]	Analysis of AXEPT [73]			482	Advanced CRC	Chemotherapy		0 (0)	49 (10.2)	20% overlapping Rx	N/A
Kichenadasse 2021 [53]	Analysis of 6 trials [33,39,48,63,66,92]			5359	Advanced CRC	Chemotherapy		3829 (71.4)	946 (17.7)	During Rx	N/A
Chu 2017 [36]	Analysis of TRIO013/LOGiC [43]			274	Advanced GEC	Chemotherapy		274 (100)	119 (43.4)	20% overlapping Rx	N/A

Abbreviations: RCT, randomised–controlled trial; PPI, proton pump inhibitor; CTLA-4, cytotoxic T-lymphocyte-associated protein 4; PD-1, programmed cell death protein 1; PDL-1, programmed death ligand 1; Rx, therapy; N/A, non-available; NSCLC, non-small cell lung cancers; RCC, renal cell carcinoma; UC, urothelial carcinoma; GEC, gastroesophageal carcinoma; CRC, colorectal cancer; Cape-based, capecitabine-based; IFL, irinotecan/leucovorin/5-FU; FOLFIRI, 5-FU/folinic acid/irinotecan; and FOLFOX, 5-FU/leucovorin/ oxaliplatin. † The details of cancer type (including the type and treatment line of immune therapy and the PDL-1 expression information) are available in Appendix A * Details of PPI are presented in Appendix A, ¶ Although both IMpower 130 and 131 did report the use of PPI in their study population, they did not separately provide the number of PPI use in patients taking ICI and chemotherapy. Thus, we noted it as N/A but the overall use of PPI was provided in post hoc analysis (Hopkins 2022). § Failing 2016, only used cohort receiving first-line therapy for survival analyses.

**Table 2 cancers-15-00284-t002:** League table of pairwise comparisons in the network for the hazard ratio (with 95% CIs) of OS (A) and PFS (B).

(A) Overall Survival			
Immune check point inhibitors (*p*-score, 1.0000)	0.76 (0.67–0.85)	0.70 (0.62–0.78)	0.64 (0.53–0.78)
0.79 (0.72–0.86)	Chemotherapy (*p*-score, 0.6664)	0.85 (0.70–1.04)	0.82 (0.76–0.89)
0.67 (0.60–0.73)	0.85 (0.76–0.94)	Immune check point inhibitors and PPI (*p*-score, 0.2016)	1.07 (0.92–1.24)
0.66 (0.60–0.72)	0.84 (0.78–0.90)	0.99 (0.89–1.09)	Chemotherapy and PPI (*p*-score, 0.1319)
(B) Progression-Free Survival			
Immune check point inhibitors (*p*-score, 0.9706)	0.84 (0.70–1.00)	0.90 (0.71–1.13)	0.72 (0.61–0.85)
0.92 (0.81–1.04)	Chemotherapy (*p*-score, 0.6958)	0.84 (0.77–0.92)	0.81 (0.64–1.03)
0.80 (0.70–0.91)	0.87 (0.80–0.94)	Chemotherapy and PPI (*p*-score, 0.3217)	0.83 (0.68–1.02)
0.71 (0.62–0.80)	0.77 (0.67–0.88)	0.89 (0.78–1.01)	Immune check point inhibitors and PPI (*p*-score, 0.0119)

Note: Treatments are ranked by their *p*-score of overall survival with the top left representing the best, whereas the bottom right represents the worst. Estimates in the upper right triangle are direct comparisons, while those in the lower left are from the network meta-analysis.

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
