# Peer review of "The Association between Baseline Proton Pump Inhibitors, Immune Checkpoint Inhibitors, and Chemotherapy: A Systematic Review with Network Meta-Analysis"

_cancers, 2022, doi:10.3390/cancers15010284_

Round 1

Reviewer 1 Report

In this comprehensive meta-analysis, the authors convincingly show statistically relevant impact of the use of proton pump inhibitors (likely through GI tract dysbiosis) on the efficacy of ICI and conventional anti-cancer therapies in a sufficiently large number of various cancer cases and types. This study will be helpful in directing clinicians in the use of proton pump inhibitors in patients undergoing anti-cancer therapies. 

Author Response

We would like to thank you for your appreciation and your time and efforts spent on reviewing our manuscript.

To improve our manuscript writing as suggested, we have invited Mr. Tim Stubbings for proofreading. 

Reviewer 2 Report

Thank you for allowing me to review this meta-analysis on the association between baseline proton pump inhibitors, immune checkpoint inhibitors and chemotherapy. 

In the immunotherapy era for cancer treatment, many efforts were made to discover potential modifiers of treatment response to immune checkpoint inhibitors (ICIs). Proton pump inhibitors (PPI) are known for inducing a dysbiosis effect on gut microbiota. 

The innovative aspect of the present work is linked to the focus on the interaction between PPI against ICIs and chemotherapy. 

The topic is interesting, and the work is well-conducted. 

The method is rigorous and reported in detail. 

I suggest inserting Table 1 in the supplementary materials.

I appreciated the network plot displayed in Figure 2. 

From the analyses, the authors find that using PPI can compromise the effectiveness of ICI, making it worse than chemotherapy. This is particularly true for advanced non-small cell lung cancer and urothelial carcinoma, less for other cancers, such as melanoma. 

 Another important finding is that PPIs do not affect capecitabine-based therapy. 

The extreme heterogeneity of the analysed cancer types (their immune profile and PDL-1 expression, first of all)  is the most critical limitation of the work. The authors correctly state this point. Undoubtedly other trials on this topic are needed.  

Author Response

We would like to thank you for your constructive advice and your time and efforts spent on reviewing our manuscript. 

Comment: I suggest inserting Table 1 in the supplementary materials.

Ans: We really appreciate your valuable input to our manuscript. We understand your concern regarding the disposition of our Table 1 as it contains much information on study characteristics. The reason why we placed it in the main text was to follow PRISMA 2020 checklist, which suggests to create a table to illustrate key study characteristics of included studies. As a result, it remains in our revised version of manuscript.